# Cognitive and mental fatigue in chronic pain: cognitive functions, emotional aspects, biomarkers and neuronal correlates—protocol for a descriptive cross-sectional study

Marika C Möller [ID],[1,2] Nils Berginström [ID],[3,4] Bijar Ghafouri [ID],[5]
Anna Holmqvist [ID],[1,2] Monika Löfgren [ID],[1,2] Love Nordin [ID],[6]
Britt-Marie Stålnacke [ID] [4]

For numbered affiliations see end of article.

**Correspondence to**
Professor Marika C Möller;
marika.moller@ki.se

## ABSTRACT

**Introduction** Chronic pain (CP) is one of the most frequently presenting conditions in health care and many patients with CP report mental fatigue and a decline in cognitive functioning. However, the underlying mechanisms are still unknown.

**Methods and analysis** This study protocol describes a cross-sectional study aimed at investigating the presence of self-rated mental fatigue, objectively measured cognitive fatigability and executive functions and their relation to other cognitive functions, inflammatory biomarkers and brain connectivity in patients with CP. We will control for pain-related factors such as pain intensity and secondary factors such as sleep disturbances and psychological well-being. Two hundred patients 18–50 years with CP will be recruited for a neuropsychological investigation at two outpatient study centres in Sweden. The patients are compared with 36 healthy controls. Of these, 36 patients and 36 controls will undergo blood sampling for inflammatory markers, and of these, 24 female patients and 22 female controls, between 18 and 45 years, will undergo an functional MRI investigation. Primary outcomes are cognitive fatigability, executive inhibition, imaging and inflammatory markers. Secondary outcomes include self-rated fatigue, verbal fluency and working memory. The study provides an approach to study fatigue and cognitive functions in CP with objective measurements and may demonstrate new models of fatigue and cognition in CP.

**Ethics and dissemination** The study has been approved by the Swedish Ethics Review Board (Dnr 2018/424-31; 2018/1235–32; 2018/2395–32; 2019–66148; 2022-02838-02). All patients gave written informed consent to participate in the study. The study findings will be disseminated through publications in journals within the fields of pain, neuropsychology and rehabilitation. Results will be spread at relevant national and international conferences, meetings and expert forums. The results will be shared with user organisations and their members as well as relevant policymakers.

## STRENGTHS AND LIMITATIONS OF THIS STUDY

⇒ A large cross-sectional multicentre study including 200 patients and 36 controls.
⇒ Studying fatigue in chronic pain with objective neuropsychological measurements as well as biomarkers and brain imaging is an innovative approach to finding underlying pathophysiological mechanisms in this condition.
⇒ The patients are consecutive clinical patients currently being evaluated for pain rehabilitation, which means that the results from the study are clinically generalisable.
⇒ A study limitation is the small size of the control group; however, the size of the control group is adapted for comparison of the results from the functional MRI examination and blood sampling.

**Trial registration number** NCT05452915.

## INTRODUCTION

Pain is one of the most frequently presenting conditions in healthcare. When pain has been present for 3 months, it is classified as chronic. In chronic pain (CP), the central nervous system (CNS) is affected by an imbalance in the ascendent and descendent tracts. This facilitates the pain and impedes pain inhibition. This, in turn, leads to plastic changes in the CNS, which are shown as increased sensitivity to pain and other stimuli, depression and fatigue.[1–3]

Fatigue is a common report across CP conditions, not least in widespread pain,[4–9] and it is often reported to be one of the most burdensome symptoms.[10] Fatigue, however, is difficult to investigate because it is subjective in nature and the definition of the concept is indistinct.[11] Fatigue is usually measured by questionnaires, but when using questionnaires, the results often

interfere with other psychological conditions, for example, depression,[12] sleep disturbances[13] or pain.[14] As depression and sleep disorders are common in patients with CP, it is difficult to determine whether the fatigue seen in these patients is primarily related to pain or whether it is secondary to depression,[15] which incites a general attitude that fatigue among patients with CP is due to sleep disorders and depression.[16] Still, fatigue can, but need not, be linked to impaired cognitive functioning.[17] Cognitive aspects of fatigue, cognitive fatigue, is a state where a cognitively demanding task gives rise to a mental/subjective experience of fatigue and has often been used synonymously with mental fatigue.[18] However, it is still a subjective experience. Some cognitive aspects of fatigue can also be measured on a functional level as cognitive fatigability, defined as performance decline on attention-demanding tasks.[19–21]

It is well established from studies with self-reported data and cognitive tests that patients with CP are severely bothered by cognitive impairment,[22–24] not least patients with generalised pain.[25] Also, there are several studies showing an association between pain and deficits in attention, memory, processing speed and executive function.[22 26] It is important to separate self-reported symptoms from functional disabilities on objective tests since, as noted above, self-reports can be influenced by psychological well-being.[27]

Impaired cognitive ability in CP has been explained either by 'limited resources' that is, that the pain competes for limited attentional resources in the brain,[25 28] or by neuroplastic changes, where impaired cognitive ability in patients with CP is supposed to be caused by neurodegenerative processes. In CP, the CNS is affected by an imbalance in the brain's pathway system, which leads to plastic changes in the brain.[1] In particular, pain activates areas of the brain that are important for attentional functions, which may explain why many patients with CP experience attention deficits.[22]

In patients with acquired brain injuries and other neurological conditions, self-rated fatigue has been associated with impaired attention[29] and slower processing speed.[30] Fatigability in cognitively demanding tasks has been shown in patients with mild traumatic brain injury,[31] multiple sclerosis[32] and hormonal conditions,[33 34] thus constituting an objective measure of cognitive fatigue. In acquired brain injuries and other neurological conditions, cognitive fatigue has been associated with dysfunctions in the corticostriatal networks in the basal ganglia.[35–38] The relation between fatigue and attention deficits in CP has not yet been pinpointed in the literature, and cognitive fatigability has, to our knowledge, not been studied in patients with CP. It cannot be ruled out that fatigue in CP and neurological conditions might share underlying mechanisms, and that objectively measured cognitive fatigability could be an objective marker of cognitive fatigue also in CP.

As fatigue is an indistinctive report with multifactorial origin, the importance of a standardised taxonomy has been emphasised to clarify and improve the assessment and reporting of fatigue.[11] Correlations between depression and self-rated fatigue are seen in several studies regardless of the underlying medical condition.[12 39 40]

Finding methods to capture fatigue without the result being affected by depression are important to disentangle other possible underlying causes of fatigue and thereby enable effective treatment. In previous studies on patients with brain injuries, cognitive fatigability has not been affected by depression.[41 42] By combining self-rating measures and fatigue measures not related to depression but rather with functional networks in the brain[31] or with results on cognitive tests,[41] a clearer picture of possible underlying causes of fatigue can be obtained and in the long run also what treatment should be recommended.

Functional MRI (fMRI) may reflect increased or decreased neuronal activity in the brain as changes in neuronal activity are linked to changes in regional blood flow and blood oxygen level dependent (BOLD). With different fMRI techniques, it is possible to study whether discrete cerebral dysfunctions can be linked to self-perceived fatigue and cognitive fatigue/fatigability on a group level. Fatigue among patients with brain injuries has been linked to specific networks for attention in the brain, involving frontosubcortical regions.[43–45] Möller and Nordin have shown that fatigue in mild traumatic brain injury is related to altered connectivity in the brain[45] and that different neural networks are related to subjective compared with objective measures of fatigue.[31] Whether similar patterns can also be seen in patients with pain and fatigue needs to be investigated. Connectivity changes have also been seen in CP, and patients with different pain conditions show connectivity changes.[46] These studies have not included fatigue and fatigability and have not used task-fMRI in the scanner.

Since chronic low-grade inflammation is thought to play a major role in the onset of CP and some previous studies have shown elevated levels of inflammatory biomarkers in patients with chronic generalised pain and neuropathic pain,[47–50] the immunological biomarkers in the present study are being investigated to improve knowledge on underlying mechanisms, which may lead to future potential diagnostic markers. A low-grade systemic inflammation may be related to fatigue and cognition as previous studies have suggested an upregulation or dysregulation of components in the immune system in patients with chronic fatigue syndrome,[51 52] and Wåhlén et al have reported significant correlations between immunity proteins and psychological distress in patients with fibromyalgia.[53] As an answer to the activation of the immune system activation in central immunocompetent cells such as microglia and astrocytes are initiated, resulting in a production of pro-inflammatory cytokines in the CNS that promotes a change in behaviour response characterised by decreased mood, increased anxiety, pain sensitivity, etc. Previous studies have investigated single inflammatory markers and reported inconsistent results.

There is, thus, a need for increased knowledge about the association between CP, cognition and fatigue and possible underlying pathophysiological mechanisms. To improve diagnostic and prognostic models, we aim to investigate the coexpression profile of inflammatory biomarkers and by

applying network analysis we aim to discover cellular pathways that might be activated in cognitive fatigue.

## Objectives

The primary aim of the study is to investigate the presence of self-rated mental fatigue, objectively measured cognitive fatigability and executive functions and how these are related to other cognitive functions, pain-related factors, inflammatory biomarkers and brain connectivity in patients with CP.

1. Do patients with CP show more pronounced self-rated mental fatigue and cognitive fatigue compared with healthy controls?
2. Are the different fatigue measures related to cognitive functions such as process speed and attention functions?
3. What is the relationship between (a) cognitive impairments, as measured by several different cognitive tests, (b) mental fatigue as measured by Multidimensional fatigue inventory-20 (MFI-20) and visual scale on fatigue (VAS-f) or (c) cognitive fatigability as measured by Wechsler Adult Intelligence Scale-III (WAIS-III) Coding and
   i. the duration of pain.
   ii. generalisation or intensity of pain
      and are they affected by covariates, such as sleep disorders and degree of depression/anxiety?
4. Is there an association between immunological biomarkers and (a) cognitive functions, (b) mental fatigue and (c) cognitive fatigability?
5. Does the connectivity in the brain of female patients with CP differ from healthy controls at rest as well as during activity while performing a vigilance task?

6. Is there an association between connectivity in the brain and
   a. immunological biomarkers.
   b. fatigability.
   c. results on neuropsychological tests?

## METHODS AND ANALYSIS

### Study design

This is a cross-sectional explorative case–control study. Strengthening the Reporting of Observational Studies in Epidemiology-reporting guidelines will be used.[54]

### Study setting

The Unit of Pain Rehabilitation at the Department of Rehabilitation Medicine at Danderyd University Hospital, Stockholm and the Department of Pain Rehabilitation, Pain Center, Umeå University Hospital, Umeå from September 2018 to October 2022. Data collection is now complete, but no analyses have been completed and reported.

### Participants

#### Patients for neuropsychological assessment

Two hundred outpatients between 18 and 50 years with CP from the Unit of Pain Rehabilitation at the Department of Rehabilitation Medicine at Danderyd University Hospital or from the Department of Pain Rehabilitation, Pain Center at Umeå University Hospital will be included. There will be no stratification of age, but the majority of the patients will be women as they are over-represented among those receiving rehabilitation for CP. In the analyses, the patients are compared with 36 healthy controls.

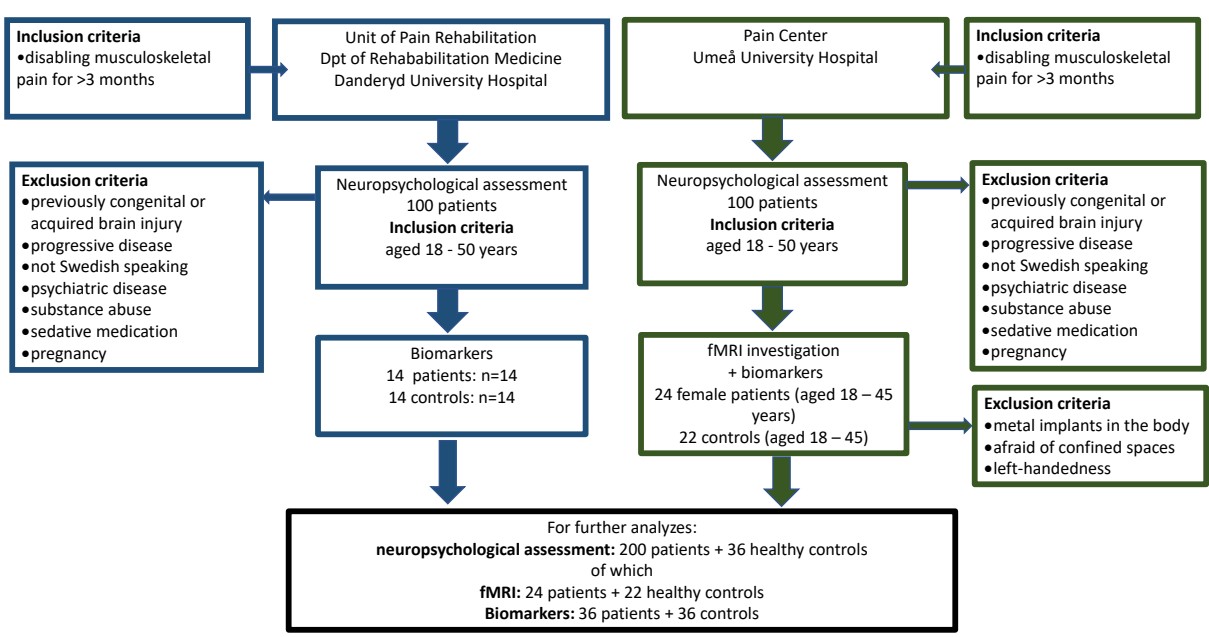

**Figure 1** Study flowchart. fMRI, functional MRI.

Please see the flowchart for an overview of the study (figure 1).

## Healthy controls

Thirty-six healthy controls will be primarily used for comparison of the results from the fMRI examination and the blood sampling. They will be matched at a group level in terms of gender, age and educational level. They will be recruited among hospital staff or their friends. For those who only undergo neuropsychological assessment and blood sampling, a token of Swedish krona (SEK) 300 will be offered and for those who do the fMRI and blood sampling, a token of SEK 1000 will be offered.

## Participants for fMRI

Twenty-four[24] patients, women only, between 18 and 45 years with CP and who have undergone a neuropsychological assessment at the Department of Pain Rehabilitation, Pain Center at Umeå University Hospital will also undergo an fMRI investigation. All patients fulfilling the criteria for participation in the fMRI study from Umeå the study site are offered to participate in fMRI scanning and blood sampling, until the participation rate goal[24] is reached. They will be compared with 22 female controls from the healthy control group.

## Participants for blood sampling

Of the included participants, 36 patients and 36 controls will undergo blood sampling for immunological markers. All participants undergoing fMRI will also be eligible for blood sampling, as stated above. In addition, 14 patients from the Danderyd site will be consecutively recruited for the blood sampling part of the study.

## Eligibility criteria
### Inclusion criteria
► Disabling musculoskeletal pain for >3 months.
► Aged 18–50 years for neuropsychological assessment and 18–45 for fMRI examination.
► Referred to the Department of Rehabilitation Medicine at Danderyd University Hospital or to the Department of Pain Rehabilitation, Pain Center, Umeå University Hospital.

### Exclusion criteria
► Traumatic brain injury (including concussion).
► Extensive psychiatric problems or substance abuse.
► Previously congenital or acquired brain injury and/or does not live in their own home and who needs support in everyday life.
► Lack of knowledge of the Swedish language.
► Progressive disease.
► Drugs with a strong sedative effect.
► Pregnancy.
► For fMRI examination, patients with metal objects implanted in the body or metal chips in body parts

(eg, after eye damage) are excluded. Patients are also excluded if they are afraid of confined spaces or if they are left handed.

## Measurements

Background information from the Swedish Quality Register for Pain Rehabilitation (SQRP) (http://www.ucr.uu.se/nrs/). Information about medication, possible trauma and background information and degree of sick leave will be obtained from the medical record.

### Questionnaires from SQRP
► EuroQoL five dimensions questionnaire[55]
► Hospital Anxiety and Depression Scale[56]
► Insomnia Severity Index[57]
► Multidimensional Pain Inventory[58]
► Screening on physical activity by the National Board of Health and Welfare.[59]

### Other questionnaires
► VAS-f before and after neuropsychological assessment[31]
► VAS-f and pain before and after the fMRI scanning
► MFI-20[60]
► Pain level, experienced during the neuropsychological examination, will be rated (0–10) at the end of the session.

### Neuropsychological tests
► WAIS-III Digit-Symbol-Coding (Coding) measures psychomotor speed and incidental memory. The higher values the better results.[61] The Coding subtest from WAIS-III will be used instead of WAIS-IV as the former has been validated for measuring fatigability.[33 42]
► WAIS-IV Matrix reasoning measures logical reasoning and premorbid cognitive level. The higher values the better results.[62]
► WAIS-IV Digit Span measures verbal attention span and working memory. The higher values the better results.[63 64]
► Delis-Kaplan Executive Function Scale Color-Word Test and Word Fluency Test measure different aspects of executive functioning. For the Color-Word test, lower values represent better results and for the Word Fluency Test, higher values represent better results.[65]
► Ruff 2 & 7 measures automatic visual scanning and complex selective attention and processing speed. The higher values the better results.[66]
► MapCog Spectra is a newly developed attention test that is administered via an iPad. The MapCog test records and analyses the occurrence and frequency of attentional lapses and compares these to standardised norms. The standardisation is based on a healthy reference group, n=313, aged 8–87 years. The test has a sensitivity of 95.8% and a specificity of 96% for identifying individuals with attention deficit disorder. The test is easy to perform (5–10 min), the

**Table 1** Outcome measures

| Primary outcome measures | Description |
|---|---|
| 1. Fatigability from WAIS-III Coding | The subject must fill in the blank spaces with the symbol which is paired to the number during 120 s. Cognitive fatigue is assessed by subtracting the number of digits produced in the first 30 s from the number of digits produced in the last 30 s during the full 120 s period. A non-ascending score (<0) is considered an indicator of cognitive fatigue. Both the total value and the difference between the production between 0–30 s and 91–120 s are measured, and a dichotomised variable (non-ascending value) will be used. |
| 2. D-KEFS Color-Word Test | Inhibition of over-learnt verbal responses. The test has four conditions: (1) naming colours (red, blue, or green, (2) reading colour words printed in black, (3) naming the colour of the words red, blue, or green written in a different colour than is the written word, which means inhibition of an over-learnt function of reading the word; (4) repeatedly switching between naming colours and reading out the printed words as quickly as possible, while at the same time, the person needs to keep track of clues that indicate rule change. Contrast scores are used to examine the performance of the more complex tasks 3 and 4 and the basic tasks 1 and 2. Faster time represents a better result. |
| 3. Fatigability on e-prime vigilance task in the fMRI scanner | The participants are instructed to push a button as quickly as they can when a set of four zeroes appears in a red rectangle and do nothing if other numbers appear. After each response, visual feedback of the reaction time is displayed. If the participant reacts to a false stimulus or if the response time exceeds 1 s the feedback 'false answer' or 'no answer' is displayed. The stimuli are presented at random intervals. The results are divided into quintiles and the mean reaction time is calculated for each quintile. |
| 4. Task-fMRI | BOLD signal changes during the e-prime vigilance task. |
| 5. Resting-state fMRI | Changes in functional connectivity after performance of the e-prime vigilance task. |
| 6. Inflammatory markers | Omics analyses will be used in this study. Omics studies are mainly exploratory and hypothesis-generating, that is, which proteins, metabolites, and lipoproteins are identified cannot be determined in advance. Exploratory targeted analyses using panels of cytokines and chemokines and neuroinflammation will be used in combination with omics. |
| Secondary outcome measures | Description |
| 1. MFI-20 | The MFI-20 consists of five scales, based on different modes of expressing fatigue. Each scale contains four items for which the person has to indicate on a seven-point scale to what extent the particular statement applies best. 'General fatigue' includes general statements concerning a person's functioning. 'Physical fatigue' refers to the physical sensation related to the feeling of tiredness. 'Reduced activities' measures reduction in activities and 'Reduced motivation' is lack of motivation. 'Mental fatigue' measures cognitive symptoms related to fatigue. Some sentences are inverted and need to be rescored. On each scale, higher values represent higher levels of fatigue. |
| 2. Visual Analog Scale of Fatigue | Measurement of self-rated current fatigue level. Ranges from 0 (corresponding to no fatigue) to 100 millimetres (corresponding to the worst possible fatigue). |
| 3. D-KEFS Word Fluency Test | The test measures expressive language skills, initiative, and working memory and consists of three different conditions: verbal phonological fluency[1] where the test person for 60 s produces as many unique words as possible that begin with a given letter; category fluency[2] where the test person for 60 s per category produces as many unique words as possible in two given semantic categories and; category change fluency[3] where the test person produces as many unique words as possible in 60 s and switches between two specified semantic categories every other time. The more words, the better performance. |
| 4. WAIS-IV Digit Span forward and backward repetition (Attention span and working memory) | The test person must repeat numbers that the leader reads out. The number of digits is increased by one unit every two times. The test person repeats the numbers in the same order (forward repetition) or reverse order (backward repetition). Forward repetition measures auditory attention span and short-term memory, while backward repetition measures auditory working memory. Both the total number of digits and the difference between forward and backward repetition are measured. |

BOLD, Blood-Oxygen-Level Dependent; Coding, Digit-Symbol Coding Test; D-KEFS, Delis-Kaplan Executive Function Scale; fMRI, functional MRI; MFI, Multidimensional Fatigue Inventory; WAIS, Wechsler Adult Intelligence Scale.

data processing is automatic, the results are objective and there are no learning effects. The lower values the better results.[67] We also measure fatigability by comparing the results at the end of the test with the beginning of the test.

For all tests, mainly raw score scores will be used in the analysis.

**Primary outcome variables**

The outcome variables are summarised in table 1.

## DATA COLLECTION AND MANAGEMENT
### Recruitment

The pain rehabilitation unit at the Department of Rehabilitation Medicine at Danderyd University Hospital and the Department of Pain Rehabilitation, Pain Center at Umeå University Hospital offer team assessment and evidence-based team-based multimodal rehabilitation for patients with complex CP (>3 months). To receive rehabilitation at the clinics, a referral including a pain assessment is required. Information about the study is sent to the referred patients at the same time as the call for the initial assessment.

At the time of the team assessment, the patient is asked orally by the team nurse if the testing psychologist may contact the patient to inform them about the study. The patient's consent is logged, and a log is made of the number who say yes and the number and gender and age of those who say no. No personal data other than gender and age will be registered for those who say no. The purpose of the log is only to be able to assess generalisability based on the proportion who have agreed to participate in the study.

The testing psychologist then calls the patients and inform them about the study and if the patient agrees, an appointment is made for neuropsychological assessment. In connection with the assessment, the psychologist ensures that they have understood the information about the study and that participation is voluntary, inquires about whether they have further questions and collects the signed consent form. Female patients who are investigated in Umeå are asked if they want to participate in the fMRI study. The patients are assessed by an independent psychologist, who does not treat the patients clinically.

The study patients will be compared with healthy controls. Healthy controls will be recruited through advertising among hospital staff. Matching on a group level will be done regarding age, gender and level of education.

### Imaging

In a 3-tesla MR scanner, conventional anatomical MRI sequences will be analysed as well as sequences for detailed anatomical assessment. We use an established measurement paradigm that has previously been used on healthy subjects[68] and patients with mild traumatic brain injury.[69]

All participants are examined with fMRI during a 20 min vigilance (reaction time) task (e-prime), where fatigue is measured by recording mean reaction time/quintile. BOLD resting-state fMRI examination is performed before and after the task. In addition, the MRI protocol will include standard clinical protocols that include high-resolution T1-weighted sequences and high-resolution T2-weighted scan (Fluid-attenuated inversion recovery (FLAIR)). All structural images are reviewed by a radiologist to screen for incidental findings within the brain. The investigation is non-invasive and takes about 60 min.

### Blood sampling

Venous blood samples (10–20 mL) are collected in P100 tubes from BD diagnostic, containing a protease inhibitor cocktail that prevents protein cleavage/degradation from each subject. The blood samples are centrifuged to remove red blood cells. The plasma fraction is transferred to a new tube, portioned and stored at −86°C until analysis at the PAINOMICS lab in Linköping, Sweden.

Plasma samples will be analysed using antibody-based methods such as multiplex technology (Meso Scale Diagnostics, Rockville, MD). Instrument, MSD Technology platform, which is a multiplex instrument, which can analyse up to 92 different substances (cytokines, chemokines, growth factors and metabolic proteins) in one go. Omics analysis consisting of proteomics, metabolomics and lipoprotein profiling will be used to identify other inflammatory markers than those analysed by the antibody-based (targeted analysis) method.

### Data management

All data materials will be recorded with a participant ID and will be unidentifiable for those who do not have access to the code key. In the statistical processing, the participants will be coded. The code key is stored separately from the data and only pseudonymised data will be shared between the two sites. Deidentified data will be electronically stored on the server at the two sites and will be deleted 10 years after the project has ended. The final data set will be available to researchers actively contributing to statistical analyses and publications. Data entry will be controlled by initial exploratory analyses, including range checks, to promote data quality. Regarding biological specimens, the samples of blood will be centrifuged and plasma will be aliquoted and stored at the Linköping University Hospital, with special notification to the biobank. The samples will be blinded before analysis. This study has an exploratory character as we use proteomic methods, and, therefore, samples (that are not thawed) will be stored for future analysis according to approved ethical application.

### Sample size

Power calculation on the primary fatigue measure, the WAIS-III Coding Test: for a mean difference between patients and controls of 2.5, and SD of 3.0, and an alpha level of 5% (two-sided) and strength of 80%, a group of 24 participants in each group is a sufficiently large sample.[70] This size also holds for the fMRI substudy as an earlier study on mild traumatic brain injury has shown that 10 patients in each group were sufficient.[45] The biomarker data will be analysed using multivariate data analysis (MVDA). Algorithms for sample size calculations using MVDA do not exist. MVDA is designed to handle a few subjects, low-to-variable ratios and multiple intercorrelated variables. Based on our previous biomarker study,[48 71] the sample size in this study will be sufficient to detect a difference between groups.

## Confidentiality

Information on participants will be handled by healthcare professionals adhering to Swedish Law ensuring confidentiality and data protection by coding individual participants' collected data. Results and data will be presented at a group level in publications, rendering the identification of individual patients impossible. All data will be stored in accordance with the General Data Protection Regulation. The data for participants from the Stockholm area will be stored in a password-protected project server at Danderyd University Hospital and for patients from the Umeå region at Umeå University. Data will not be accessed by unauthorised persons.

## Data analysis

For all study questions, descriptive statistics will be used to depict demographics, injury characteristics, results on neuropsychological tests and psychological screening instruments. Multiple linear and logistic regression will be used to detect any interaction effects. Parametric methods such as independent t test, one-sample t test (comparing with standardised test norms), Pearson correlation and multiple linear regression) will be used when data are normally distributed and non-parametric methods (Mann-Whitney U-test, Spearman correlation, $\chi^2$, logistic regression) when data are skewed or on interval level or lower. Missing data will be listed as data loss. The significance level will be set to <0.05 (two-tailed).

Concerning study question 4, data from omics and targeted analysis will be analysed using advanced MVDA using SIMCA-P+. This is the recommended method in omics since it accounts for multicollinearity problems and missing data when the number of variables exceeds the number of observations.[72] MVDA will be used for the analysis of the biomarkers. Principal component analysis (PCA), orthogonal partial least squares- discriminant analysis (OPLS-DA) and OPLS regression will be applied for analysis of the biomarkers and their ability to differentiate between groups and their correlation with clinical characteristics and outcomes. PCA is an unsupervised analysis that reduces the dimensionality of data and extracts and displays systematic variation in the data matrix. Outliers will be identified using score plots in combination with Hotelling's T2 test (the distance from the origin in the model plane for each selected observation) and DModX (the distance of the observation to the X model plane). The values that are larger than the 95% confidence limit are considered suspicious and the values larger than the 99% limit should be considered strong outliers.[73]

OPLS-DA will be used to identify biomarkers responsible for group separation. The OPLS-DA model reveals variables as loadings and the higher the value of a loading, the more important it is for the model, for example, group separation. This can be measured as a variable influence on projection (VIP) values.

The p(corr) is the loading of each variable scaled as a correlation coefficient (ranging from −1 to +1). An absolute p(corr) of >0.4–0.5 and VIP >1.0 are considered significant.[72] Another model diagnosis parameters that are considered are R2 value—which describes the 'goodness of fit', that is, the fraction of the sum of squares of all the variables explained by a principal component. The Q2 value describes the 'goodness of prediction', that is, the fraction of the total variation of the variables that can be predicted by a principal component using a cross-validation method[73] and cross-validated analysis of variance (CV-ANOVA). The R2 must not differ more than 0.2–0.3 compared with Q2 and CV-ANOVA <0.05 is considered a significant model.

Regarding study question 5 analysis of fMRI data will be done with special software for MRI analysis, developed at Oxford University (http://www.fmrib.ox.ac.uk/fsl/). The resting state data will be analysed using FSL (http://www.fmrib.ox.ac.uk/fsl) tools such as MELODIC to identify spatial and temporal components by independent component analysis at the group level. All relevant preprocessing steps will be performed, such as motion correction, registration to a standard template and filtering of low-frequency drifts. Group differences between patients and controls will be analysed using FSL dual regression. The fMRI data will be analysed using both MELODIC and FEAT, which is an FSL tool based on general linear modelling performing multiple regression on the fMRI data and the functional paradigm.

For study question 6, multiple linear and logistic regression will be used to detect any interaction effects.

## Plans for communicating important protocol amendments to relevant parties

Important protocol modifications will be reported to the Ethics committee in Sweden and amendments will be made to the trial registry (Clinicaltrials.gov).

## Patient and public involvement

None.

## ETHICS AND DISSEMINATION

The study was approved by the Swedish Ethical Review Authority (2018/424-31; 2018/1235–32; 2018/2395–32; 2019–66148; 2022-02838-02). All participants are provided oral and written information by the neuropsychologists (AH and NB) and will give written informed consent to participate in the study.

Publications are planned for journals in the fields of pain, neuropsychology and rehabilitation. Results will further be spread at relevant conferences, national and international meetings and expert forums. The results will be shared with user organisations and its members as well as relevant policymakers.

**Author affiliations**
[1]Clinical Sciences, Karolinska Institutet, Stockholm, Sweden

²Rehabilitation Medicine, Danderyd University Hospital, Stockholm, Sweden
³Psychology, Umeå Universitet, Umeå, Sweden
⁴Community Medicine and Rehabilitation, Rehabilitation Medicine, Umeå Universitet, Umeå, Sweden
⁵Health, Medicine and Caring Sciences, Linköping University, Linköping, Sweden
⁶Neurobiology, Caring Sciences and Society, Karolinska Institutet, Stockholm, Sweden

**Contributors** MCM, ML and B-MS formed the original research concept. BG, LN, NB and AH contributed to the study design within their field of expertise. MCM and B-MS will coordinate the project in Stockholm and Umeå in cooperation with NB, BG, AH, ML and LN. NB and AH will be responsible for data collection and statistical analyses of the neuropsychological and demographic data in collaboration with MCM, ML and B-MS. LN will be responsible for fMRI analyses in collaboration with NB. BG will be responsible for the analyses of biomarkers. MCM has written the manuscript draft. All authors have contributed with important intellectual content to the manuscript. Regarding upcoming manuscripts, we will use authorship eligibility guidelines. We will not use professional writers.

**Funding** This study is supported by Promobilia Foundation, grant number [A22056], the research and development fund granted by the County Council of Västerbotten [RV-928951; RV-939460], and through a regional agreement between Umeå University and Västerbotten County Council (ALF) [RV-930845; RV-940388; RV-967132]. The Department of Rehabilitation Medicine at Danderyd University Hospital and the Department of Clinical Sciences, Karolinska Institutet have also provided valuable support for the study to be feasible. The funders have no vested interest in the study and have not influenced the design.

**Competing interests** None declared.

**Patient and public involvement** Patients and/or the public were not involved in the design, or conduct, or reporting, or dissemination plans of this research.

**Patient consent for publication** Not applicable.

**Provenance and peer review** Not commissioned; externally peer reviewed.

**ORCID iDs**
Marika C Möller http://orcid.org/0000-0001-8700-5186
Nils Berginström http://orcid.org/0000-0003-1192-4527
Bijar Ghafouri http://orcid.org/0000-0002-6396-5104
Anna Holmqvist http://orcid.org/0000-0001-9058-7463
Monika Löfgren http://orcid.org/0000-0002-8701-0206
Love Nordin http://orcid.org/0000-0002-9685-8583
Britt-Marie Stålnacke http://orcid.org/0000-0002-2916-0628

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
