## [Reviewer comments · BMJ Open]

ARTICLE DETAILS

TITLE (PROVISIONAL)	Cognitive and mental fatigue in chronic pain—cognitive functions, emotional aspects, biomarkers, and neuronal correlates: protocol for a descriptive cross-sectional study
AUTHORS	Möller, Marika; Berginstrom, Nils; Ghafouri, B; Holmqvist, Anna; Lofgren, Monika; Nordin, Love; Stålnacke, Britt-Marie

VERSION 1 – REVIEW

REVIEWER	Moore, David Liverpool John Moores University
REVIEW RETURNED	13-Oct-2022

GENERAL COMMENTS	The purpose of this protocol is to examine the cognitive abilities and fatigue of patients living with chronic pain and the neural correlates of these functional outcomes. To achieve this the authors propose an ambitious combination of self-report, neuropsychological tests and fMRI techniques. I do however have a number of questions about the protocol as described which I would like to ask the authors to consider. 1) The authors report an intended control group size of 36, they acknowledge the limitation of this, why not recruit more? It would seem easier to increase the power of this group and at least get closer to parity2) It is also unclear why the authors intend to have a sex imbalance between the pain and control groups for the fMRI phase, I understand that once data collection is started this might happen however it seems that one would plan to balance their groups.3) The authors state that work talking about cognitive function in chronic pain is largely based on self-report, there is a lot of work looking at performance on both standard neuropsychology tasks as well as looking at other aspects of cognitive function. these data should in my opinion be acknowledged and discussed.4) I am a little concerned about how the authors propose to use fMRI data in this study, these findings are correlational and I am not sure how the authors propose to find the causal mechanisms around fatigability here, it would be good for this to be more clearly explained.5) The authors report no PPI, could they expand on why this was not felt necessary? This might help to provide directions to how fatigue or aspects of cognition that might be especially affected by their pain.6) Compensation for time is reported for the control group but not the patient group. Can the authors confirm that this will be given to all participants?
--

	7) I am unclear why in the materials a mix of WAIS III & IV items being used? 8) RUFF 2&7 and the MapCog Spectra tasks are not standard measure to my knowledge. To be able to review the appropriateness of these they should be clearly described for the reader 9) For some of the WAIS items cognitive fatigue is defined by reduction in performance from first 30seconds to last 30seconds on a 90 second task. Will fatigue set in for such a short duration. Also why are these tasks particularly suitable? Why not a continuous performance task or another task designed more clearly for this purpose?
--	---

REVIEWER	Alexander, Caroline Curtin University, School of Occupational Therapy, Sociak Work and Speech Pathology
REVIEW RETURNED	15-Dec-2022

GENERAL COMMENTS	This is an interesting body of work from Moller and colleagues that has potential to inform future interventions for cognitive function and fatigue for patients with Chronic Pain. I suggest some significant modifications to the manuscript and seek a number of clarifications prior to this being published. Title: It may be changed to “Cognitive Fatigue” or “Mental Fatigue” as the term fatigue could include physical or general fatigue. Abstract: The abstract is clear and well written. Introduction: Several terms are used in the paper relating to “fatigue” including fatigue, cognitive fatigue, mental fatigue and fatiguability, and I was unclear if they are used synonymously or to describe different concepts. I suggest that the authors define what they mean by “fatigue” and other variations in the beginning of the introduction and remain consistent with how they use the terms throughout the paper. The introduction covered the relevant background and literature, though was at times difficult to follow and I recommend review to ensure the key points are clear and concise. It may be useful to present the evidence of fatigue in patients with CP before detailing the limitations of subjective estimates for example. Page 4, paragraph starting at line 50: References are required for the theories described. This paragraph was difficult to follow as a reader as to how the various evidence related to the theories described and the implications for your research objectives and hypotheses. This paragraph would be assisted by a clear topic sentence and summary sentence. Page 5, line 13-14: “Only a few studies” plural, however there is only one reference – is there more than one study? As you are measuring cognitive function, it would be useful to know what the study/studies found. Page 5, line 16-19: This reads as you suggesting fatiguability on cognitive tasks could be a marker to distinguish the different causes of fatigue from each other, but this does not make sense to me on face value. The remainder of the paragraph reads equally as confused as to what the reader is expected to take away from it. Page 5, paragraph starting line 28: It is unclear from the introduction if the inclusion of fMRI and investigation of brain connectivity is to provide supporting evidence for the theory of impaired cognition or to be used as a diagnostic marker.
---

	Page 5, paragraph starting line 47: Similarly I am unsure from the structure of the introduction if immunological biomarkers are being investigated as a potential underlying mechanism, or as a diagnostic marker. Also why do you expect to find a relationship between immunological biomarkers and fatigue or cognition? The proposed project is very comprehensive, and it would be worth highlighting the significance and potential application of the knowledge. Objectives: There a large number of objectives, though they appear relevant and appropriate given the background literature presented. I recommend they are reviewed to ensure they are specific and clear Objective 1: I suggest these should be two objectives: With the first being “Do patients with CP show more pronounced self-rated mental fatigue and cognitive fatigue compared to health controls?” and the second being “Are fatigue measures related to cognitive functions...”. Objective 1: Please clarify if mental fatigue and cognitive fatigue the same thing or different? Objective 1: Clarify if you are referring to self report or objective measures of fatigue in second part of the objective stating “Are fatigue measures..” Objective 2: This objective would be clearer when you clarify in the introduction the difference between mental fatigue and fatiguability. Objective 2: It can be useful to state [variable], as measured by [assessment] to ensure it is specific and clear. Objective 3: When you state cognitive functions/mental fatigue/cognitive fatiguability can you please clarify if this a list of different variables you will be looking for associations with? Or are they being used interchangeably? If the later, I advise to define your terms in the introduction and remain consistent throughout. Objective 4: I recommend you include in your objective “females with CP” as you are only conducting fMRI on females. Methods: Study setting: Has the recruitment already started? Please include the date range in which you will be recruiting. Patient and public involvement: Is there a reason consumers have not been involved in this project? Are there any plans to include consumer involvement at any stage of the project? Participants The authors made a good attempt to describe a very complex participant group as there are a number of small sub studies within the broader project. A flow chart for the eligibility, selection and inclusion of participant in the various arms of the study would be very useful. Healthy controls: It is unclear which group the 36 healthy controls will be matched to – the 200 patients for neuropsychological assessment or the 24 patients for fMRI or the 36 patients for blood sampling? How will you ensure they are an appropriate comparison group for all patient samples? Please clarify how you are selecting the patients for the fMRI. Are they randomly selected from the eligible population? Similarly how are you selecting patients for the blood sampling? It would be valuable to explain why you are only selecting female patients for the fMRI.
--	---

	You state that all patients who will have fMRI will have blood sampling, therefore at least 24/36 will be female. Please comment on the implications of this for your analysis. How will you match your controls to this group of 36? Inclusion criteria – I recommend clarifying your criteria of “CP” – do they require a formal diagnosis and made by whom? Measurements: The list of measures is fairly extensive, with most appropriately referenced. However there is no reference for VAS-f – please clarify if this is a new/adapted measure proposed, or include the relevant reference. For all neuropsychological tests it would be valuable to elaborate briefly on the assessment results and interpretation (eg standardised scores?, interpreted against normative sample?) for those not familiar with the measures. The study will use two different versions of the Wechsler Adult Intelligence Scale: the older WAIS-III and newer WAIS-IV. Can you please justify why are you using the WAIS-III for digit-symbol-coding but WAIS-IV for other measures? I am not familiar with the Ruff 2&7 and would value more comprehensive description of test and results. Similarly with the MapCog Spectra. Table 1: Outcome measures. The table would be clarified by being consistent with terminology. Is the fatiguing attention task in row 4 (Task fMRI) the same as the task in row 3, and the fatiguing vigilance task in row 5? In Row 4 of the secondary outcome measures should this refer to the WAIS-IV Digit Span? The left hand column suggests you are just looking at backward repetition, but the right hand column also refers to forward repetition. I recommend revising for clarity here, and justifying why you are selecting Data management – please clarify if the data will be unidentifiable or reidentifiable? Sample Size- It is unclear which sample the power calculation is for, and the basis for using WAIS-III Coding test as each objective has different dependent variables? The power calculation states that 24 is sufficient to detect a difference of 2.5 however, the sample sizes detailed in the participants section are 200 (neuropsychological assessments), 24 (fMRI) and 36 (blood sampling). Please justify the larger sample sizes for the first and third study component. Data analysis On surface level the statistical plan seems reasonable however it is lacking in detail. Here I would expect to see a clear link between each objective listed, and an appropriate analysis planned, and recommend careful review. Closing remarks: This protocol outlines a comprehensive body of work from Moller and colleagues on investigation into cognitive function and fatigue for patients with Chronic Pain. I recommend structural revisions to the introduction to improve the clarity and inclusion of additional details in the methods section to clarify the details and rationale.
--	---

REVIEWER	Carlson, David VA Greater Los Angeles Healthcare System, Psychiatry
REVIEW RETURNED	18-Dec-2022

GENERAL COMMENTS	This is a protocol publication of a multi-component assessment of biometric, psychological, and neurophysiological variables in patients with chronic pain compared to normal controls.  - The study limitations are well-considered and comprehensive, and the reasons for the predominantly female sample is clearly laid out - The references are appropriate and up-to-date, adequately supporting the context and relevance of the present study - The psychological tests and scales used in the protocol are appropriate for the variables being studied, and many of the Swedish language instruments are described sufficiently to allow similar instruments to be used in other languages - The methods are clearly and appropriately described This review proposes only minor revisions to the submitted document:  - One concern exists regarding the exclusion criterion of "abuse." It would help to clarify the type(s) of abuse being referred to, as physical abuse (instead of emotional abuse or other forms) could have very different effects on the outcome variables being studied and complicate reproducibility of this study. The article is well-written in academic English, with only a few minor revisions to the grammar/spelling:  - The first line of the Abstract section should say "...presenting symptoms in health care and many patients with CP report..." - The plural of analysis is "analyses" ("analyzes" is only used in American English for the third-person verb - e.g. "Thomas analyzes the information") - put a comma after "of these" when it appears (e.g. "of these, 24 patients....") - use "sensitivity to pain" instead of "sensibility for pain" - under "Patients for Neuropsychological Assessment" section, write "... with CP from the Department of Rehabilitation Medicine at [name of institution] or the Department of Pain Rehabilitation at [Name of institution]..." ---- unless you're referring to inpatients who are "admitted to the..." - The section after "Confidentiality" should be called "Data Analysis" Best regards and thank you for submitting this interesting and novel study protocol for review.
--

VERSION 1 – AUTHOR RESPONSE

Reviewer: 1
Dr. David Moore, Liverpool John Moores University

Dear Dr Moore,
Thank you for taking your valuable time to read and comment on our manuscript so carefully. Your comments have contributed to making the manuscript clearer and more rigorous. We have answered

your comments point by point below and when necessary, rephrased the manuscript accordingly. We now hope that you think the study protocol is clearer and can be accepted for publication.

Comments to the Author:

The purpose of this protocol is to examine the cognitive abilities and fatigue of patients living with chronic pain and the neural correlates of these functional outcomes. To achieve this the authors propose an ambitious combination of self-report, neuropsychological tests and fMRI techniques. I do however have a number of questions about the protocol as described which I would like to ask the authors to consider.

1) The authors report an intended control group size of 36, they acknowledge the limitation of this, why not recruit more? It would seem easier to increase the power of this group and at least get closer to parity

Response: The control group is small because it is primarily used to be able to compare the results from the fMRI examination and the blood sampling. For the neuropsychological results, we can compare with standardized test norms. We have clarified this in the text, page 8, lines 140- 143

2) It is also unclear why the authors intend to have a sex imbalance between the pain and control groups for the fMRI phase, I understand that once data collection is started this might happen however it seems that one would plan to balance their groups.

Response: Our ambition was to include consecutive patients, but since about 90% of the patients are women, so few men will only give uncertainty in the fMRI analysis. It was therefore more rational to include only women in the fMRI part of the study. Left-handed people are also usually avoided because functional brain functions can be different in left-handed people. Hence, it is customary in fMRI studies to include only right-handed people. The same thought was thus the reason for examining only women and not affecting the purity of the study with a few men.

3) The authors state that work talking about cognitive function in chronic pain is largely based on self-report, there is a lot of work looking at performance on both standard neuropsychology tasks as well as looking at other aspects of cognitive function. these data should in my opinion be acknowledged and discussed.

Response: Thanks for pointing out this ambiguity in the script. Admittedly, self-reports are more common, but you are right that there have also been many studies that included cognitive tests. We have now partly rewritten the introduction and added more references, taken into consideration your recommendations. For this comment please see page 4, lines 23-24.

4) I am a little concerned about how the authors propose to use fMRI data in this study, these findings are correlational and I am not sure how the authors propose to find the causal mechanisms around fatigability here, it would be good for this to be more clearly explained.

Response: Your remark is absolutely valid that fMRI findings are correlational, hence we cannot draw causal inferences from the results. However, as stated in the title, and in the objectives section, we are searching for “neuronal correlates” (title) and associations and relationships (objectives). We have also reformulated our introduction and hopefully this is clearer now. Page 6, lines 94-96.

5) The authors report no PPI, could they expand on why this was not felt necessary? This might help to provide directions to how fatigue or aspects of cognition that might be especially affected by their pain.

Response: This is certainly an interesting suggestion from the reviewer. However, since there have been very few studies focusing on fatigue in patients with chronic pain, we prefer to do data-driven analysis in a more explorative manner.

6) Compensation for time is reported for the control group but not the patient group. Can the authors confirm that this will be given to all participants?

Response: As the patients are clinical patients who have sought care for their symptoms and complaints, the supplementary investigations carried out are believed to be of interest to the patient her/himself. As the participation is voluntary and they receive feedback on their test results and therefore may benefit clinically from the examinations, we have chosen not to reimburse them for the time they spent participating in the study.

7) I am unclear why in the materials a mix of WAIS III & IV items being used?

Response: Only the Coding test is from the WAIS-III battery. Other WAIS tests are from the WAIS-IV. The fact that we chose to use the Coding test from the WAIS-III battery is because that particular version of the test has been validated for fatigability in previous studies for other patient groups. See Moller et al, 2014. We have clarified the reason for this in the manuscript Pages 9-10. Lines 197-199

8) RUFF 2&7 and the MapCog Spectra tasks are not standard measure to my knowledge. To be able to review the appropriateness of these they should be clearly described for the reader

Response: While Ruff 2&7 is an established neuropsychological measure, please see the manual (Ruff RM, Allen CC. Ruff 2 & 7 Selective Attention Test. Lutz: Psychological Assessment Resources, Inc.; 1996), this, as you noted, is not the case with Mapcog Spectra. Thank you for noticing that. We have now added a more detailed description in the list of neuropsychological tests in the methods section. Page 10 , lines 209-216.

9) For some of the WAIS items cognitive fatigue is defined by reduction in performance from first 30 seconds to last 30 seconds on a 90 second task. Will fatigue set in for such a short duration. Also why are these tasks particularly suitable? Why not a continuous performance task or another task designed more clearly for this purpose?

Response: The duration of the WAIS -III subtest is 120 seconds, not 90 seconds. This test has been used to assess cognitive fatigability in other patient groups e.g., traumatic brain injury, and the results indicate that fatigability can be induced during a short task if it is cognitively demanding or requires simultaneous processing between different cognitive domains. As the patients undergo a comprehensive neuropsychological examination, we did not want to subject them to a long vigilance test. Several studies have shown that fatigue can be captured even with shorter tests if they are sufficiently mentally demanding. However, one cannot of course rule out that other networks are involved in fatigue on a vigilance test compared to fatigue on a test that requires the involvement of executive networks.

(For further reading please see: Möller, M. C., Nygren de Boussard, C., Oldenburg, C., & Bartfai, A. (2014). An investigation of attention, executive, and psychomotor aspects of cognitive fatigability. *J Clin and Exp Neurops*, 26, 1-14;

DeLuca, J. (2005). Fatigue, Cognition, and Mental Effort. In J. DeLuca (Ed.), *Fatigue as a window to the brain* (pp. 37-57). Cambridge: MIT Press and Walker, L. A. S., et al. (2019). Cognitive Fatigability Interventions in Neurological Conditions: A Systematic Review." *Neurol Ther* 8(2): 251-271).

We have clarified the reason why we chose WAIS-III for Coding in the methods section in Pages 9-10. Lines 197-199.

Reviewer: 2

Dr. Caroline Alexander, Curtin University

Dear Dr Alexander,

Thank you for taking your valuable time to read and comment on our manuscript so carefully. Your comments have contributed to making the manuscript clearer and more rigorous. We have answered your comments point by point below and have changed the script as best we could. We now hope that you feel that the study protocol is clearer and can be accepted for publication

Comments to the Author:

This is an interesting body of work from Moller and colleagues that has potential to inform future interventions for cognitive function and fatigue for patients with Chronic Pain. I suggest some significant modifications to the manuscript and seek a number of clarifications prior to this being published.

1) Title: It may be changed to “Cognitive Fatigue” or “Mental Fatigue” as the term fatigue could include physical or general fatigue.

Response: We agree that a change of title would increase clarity of the project. As we measure both cognitive and mental fatigue, we have chosen to include both of these terms in the title. Cognitive and Mental Fatigue in Chronic Pain - Cognitive Functions, Emotional Aspects, Biomarkers and Neuronal Correlates protocol for a Descriptive Cross-sectional Study. Note that the title also has been changed according to the instructions from the editor.

Abstract: The abstract is clear and well written.

Introduction:

2) Several terms are used in the paper relating to “fatigue” including fatigue, cognitive fatigue, mental fatigue and fatiguability, and I was unclear if they are used synonymously or to describe different concepts. I suggest that the authors define what they mean by “fatigue” and other variations in the beginning of the introduction and remain consistent with how they use the terms throughout the paper.

Response: We agree that it can be difficult to sort out what is meant in the introduction as fatigue is unfortunately not a uniform concept and previous research has not been so accurate with the taxonomy. This makes it confusing when previous studies are reported. Mental fatigue, cognitive fatigue and fatigability are different concepts. We have rewritten the introduction and tried to clarify where it is possible. We hope that this has elucidated what we aim to study. Pages 4-6.

3) The introduction covered the relevant background and literature, though was at times difficult to follow and I recommend review to ensure the key points are clear and concise. It may be useful to present the evidence of fatigue in patients with CP before detailing the limitations of subjective estimates for example.

Response: We have now rewritten the introduction, in line with your recommendations and hope that this has made the key points clearer and more concise. Pages 4-6.

4) Page 4, paragraph starting at line 50: References are required for the theories described. This paragraph was difficult to follow as a reader as to how the various evidence related to the theories described and the implications for your research objectives and hypotheses. This paragraph would be assisted by a clear topic sentence and summary sentence.

Response: We have now reviewed the whole introduction section and hope it is easier to follow. Pages 4-6.

5) Page 5, line 13-14: “Only a few studies” plural, however there is only one reference – is there more than one study? As you are measuring cognitive function, it would be useful to know what the study/studies found.

Response: Thank you for pointing this out. We have now reviewed the whole introduction section, taken into consideration your recommendations and added more references. Pages 4-6.

6) Page 5, line 16-19: This reads as you suggesting fatiguability on cognitive tasks could be a marker to distinguish the different causes of fatigue from each other, but this does not make sense to me on face value. The remainder of the paragraph reads equally as confused as to what the reader is expected to take away from it.

Response: We agree with you that this paragraph was confusing. After reviewing the introduction section, we hope that our text is now clearer and easier to follow. Page 5, lines 50-59.

7) Page 5, paragraph starting line 28: It is unclear from the introduction if the inclusion of fMRI and investigation of brain connectivity is to provide supporting evidence for the theory of impaired cognition or to be used as a diagnostic marker.

Response: Thank you for pointing out that we expressed ourselves unclearly in the introduction. Since we analyze at the group level, we cannot draw conclusions at the individual level. fMRI can thus not be used as a diagnostic marker. When we have rewritten the introduction, we have changed the wording to emphasize the importance of objective measures instead of markers to avoid misunderstandings. We have also included on group level in the paragraph page 5 line 64.

8) Page 5, paragraph starting line 47: Similarly I am unsure from the structure of the introduction if immunological biomarkers are being investigated as a potential underlying mechanism, or as a diagnostic marker. Also why do you expect to find a relationship between immunological biomarkers and fatigue or cognition?

The proposed project is very comprehensive, and it would be worth highlighting the significance and potential application of the knowledge.

Response: We have rewritten the paragraph to highlighting the significance and potential application of the knowledge as follows "Since chronic low-grade inflammation is thought to play a major role in the onset of chronic pain and some previous studies have shown elevated levels of inflammatory biomarkers in patients with chronic generalized pain and neuropathic pain [ref], the immunological biomarkers in the present study are being investigated to improved knowledge in underlying mechanism which may lead to future potential probable diagnostic markers. A low-grade systemic inflammation may be related to fatigue and cognition as previous studies have suggested an upregulation or dysregulation of components in the immune system in patients with chronic fatigue syndrome [ref] and Wåhlén et al have reported significant correlation between immunity proteins and psychological distress in patients with fibromyalgia [ref]. As an answer to the activation of the immune system initiate activation in central immunocompetent cells such as microglia and astrocytes, resulting in a production of pro-inflammatory cytokines in the CNS that promote a change in behavior response characterized by decreased mood, increased anxiety and pain sensitivity etc. Previous studies have investigated single inflammatory markers and reported inconsistent results." page 6 lines 74 -86

Objectives: There a large number of objectives, though they appear relevant and appropriate given the background literature presented. I recommend they are reviewed to ensure they are specific and clear

9) Objective 1: I suggest these should be two objectives: With the first being "Do patients with CP show more pronounced self-rated mental fatigue and cognitive fatigue compared to health controls?" and the second being "Are fatigue measures related to cognitive functions...".

Response: We have now divided the first objective into two separate objectives. Page 6 lines 97-100.

10) Objective 1: Please clarify if mental fatigue and cognitive fatigue the same thing or different?

Response: We agree that it can be difficult to sort out what is meant in the introduction as fatigue is unfortunately not a uniform concept and previous research has not been so accurate with the taxonomy. This makes it confusing when previous studies are reported. Mental fatigue, cognitive

fatigue and fatigability are different concepts. We have rewritten the introduction and tried to clarify where it is possible. We hope that this has elucidated what we aim to study. Pages 4-6.

11) Objective 1: Clarify if you are referring to self report or objective measures of fatigue in second part of the objective stating “Are fatigue measures..”

Response: We refer to both self-reported fatigue measures and objective measures of fatigability and have rephrased the sentence as follows ...and are the different fatigue measures related to cognitive functions such as process speed and attention functions. Page 6 lines 79 – 80.

12) Objective 2: This objective would be clearer when you clarify in the introduction the difference between mental fatigue and fatiguability.

Response: We have clarified in the introduction and now hope that this has become clearer and we also include cognitive fatigability (now objective 3) to clarify what we mean. Page 7 line 103.

13) Objective 2: It can be useful to state [variable], as measured by [assessment] to ensure it is specific and clear.

Response: As we divided Objective 1 to two objectives – this is now Objective 3. We have now stated specific assessments in the sentence. However, as it can affect readability to state so many cognitive assessments, we have chosen to write several different cognitive tests to facilitate readability. Page 7 lines 101-103.

14) Objective 3: When you state cognitive functions/mental fatigue/cognitive fatiguability can you please clarify if this a list of different variables you will be looking for associations with? Or are they being used interchangeably? If the later, I advise to define your terms in the introduction and remain consistent throughout.

Response: We hope that this has become clearer now that we have rewritten the introduction, but we have also made it clear that these are different measures by e.g. write a) cognitive impairments, b) mental fatigue, or c) cognitive fatigability Page 7 lines 107-108.

15) Objective 4: I recommend you include in your objective “females with CP” as you are only conducting fMRI on females.

Response: Thank you for pointing this out. We have included “females” in the sentence. Page 7, line 109

Methods:

16) Study setting: Has the recruitment already started? Please include the date range in which you will be recruiting.

Response: Response: We have now included this information in the section Study setting. Page 7, line 123 – 124.

17) Patient and public involvement: Is there a reason consumers have not been involved in this project? Are there any plans to include consumer involvement at any stage of the project?

Response: No, there was no reason for this, but we recognize that it could have been an advantage to include patients when planning future studies.

Participants

18) The authors made a good attempt to describe a very complex participant group as there are a number of small sub studies within the broader project. A flow chart for the eligibility, selection and inclusion of participant in the various arms of the study would be very useful.

Response: Thank you for pointing this out. We have included a flow chart, which will be included in the paper. Page 8, line 140.

19) Healthy controls: It is unclear which group the 36 healthy controls will be matched to – the 200 patients for neuropsychological assessment or the 24 patients for fMRI or the 36 patients for blood sampling? How will you ensure they are an appropriate comparison group for all patient samples?

Response: The control group is small because it is primarily used to be able to compare the results from the fMRI examination and the blood sampling. For the neuropsychological results, we can compare with standardized test norms, if needed and we can also control for any group differences statistically. We have clarified this in the text, page 8, lines 142 – 143.

20) Please clarify how you are selecting the patients for the fMRI. Are they randomly selected from the eligible population?

Response: Thank you for pointing out this omission in the manuscript, and it has now been clarified under heading Participants for fMRI: “All patients fulfilling the criteria for participation in the fMRI study from Umeå study site are offered to participate in fMRI scanning and blood sampling, until the participation rate goal (24) is reached.” Page 8 lines 150-152.

21) Similarly how are you selecting patients for the blood sampling?

Response: All participants undergoing fMRI will also be eligible for blood sampling, as stated above. In addition, 14 patients from the Danderyd site will be consecutively recruited to the blood-sampling part of the study. We have adjusted the text in the manuscript accordingly. Page 8, lines 156-158.

22) It would be valuable to explain why you are only selecting female patients for the fMRI.

Response: Our ambition was to include consecutive patients, but since about 90% of the patients are women, so few men will only give uncertainty in the fMRI analysis. It was therefore more rational to include only women in the fMRI part of the study. Left-handed people are also usually avoided because functional brain functions can be different in left-handed people. Hence, it is customary in fMRI studies to include only right-handed people. The same thought was thus the reason for examining only women and not affecting the purity of the study with a few men.

23) You state that all patients who will have fMRI will have blood sampling, therefore at least 24/36 will be female. Please comment on the implications of this for your analysis. How will you match your controls to this group of 36?

Response:

Thank you for this valuable comment! We agree that it is important with a sex matched healthy controls when analyzing biomarkers. Because women are over-represented among the patients, the mostly women will undertake blood sampling. The healthy controls will be sex matched. As previously we have reported [ref *], the biomarkers analysis will consider sex as confounding factor and the results will be adjusted for sex using advanced multivariate statistical analysis. Depending on which protein/proteins are analyzed the effect of sex is different therefore the results will be adjusted for sex differences when it is necessary.

Most likely we will only have a few single men who underwent blood sampling, and this is unlikely to affect the results, but we can control for this with statistical analyses. We have commented on that in the paper. Page 13 lines 317 - 339.

* Backryd, E., et al., Multivariate proteomic analysis of the cerebrospinal fluid of patients with peripheral neuropathic pain and healthy controls - a hypothesis-generating pilot study. *J Pain Res*, 2015. 8: p. 321-33. Jönsson, M. et. al. Differences in plasma lipoprotein profiles between patients with chronic peripheral neuropathic pain and healthy controls: an exploratory pilot study. *PAIN Reports* 7(5):p e1036, September/October 2022.

24) Inclusion criteria – I recommend clarifying your criteria of “CP” – do they require a formal diagnosis and made by whom?

Response: Inclusion criteria was CP med disabling musculoskeletal pain for >3 months. This information is added on page 8, line 162.

Disabling chronic pain is also an inclusion criterion for admission to the Pain Rehabilitation Programmes. The referring physician, usually in Primary Care, has examined the patient and assessed the pain when doing the referral to the two Departments of Rehabilitation.

Measurements:

25) The list of measures is fairly extensive, with most appropriately referenced. However there is no reference for VAS-f – please clarify if this is a new/adapted measure proposed, or include the relevant reference.

Response: Measuring fatigue with the VAS scale has been used in a previous publication by Möller and Nordin in the following study: Möller MC, Nordin LE, Bartfai A, Julin P, Li TQ. Fatigue and Cognitive Fatigability in Mild Traumatic Brain Injury are Correlated with Altered Neural Activity during Vigilance Test Performance. *Front Neurol.* 2017;8:496. We have included that reference in the list. Page 9, line 190.

26) For all neuropsychological tests it would be valuable to elaborate briefly on the assessment results and interpretation (eg standardised scores?, interpreted against normative sample?) for those not familiar with the measures.

Response: We have now clarified this with a line at the end of the list of neuropsychological tests as follows: “For all test mainly raw scores will be used in the analysis.” Page 10, line 218.

27) The study will use two different versions of the Wechsler Adult Intelligence Scale: the older WAIS-III and newer WAIS-IV. Can you please justify why are you using the WAIS-III for digit-symbol-coding but WAIS-IV for other measures?

Response: Only the Coding test is from the WAIS-III battery. Other WAIS tests are from the WAIS-IV. The fact that we chose to use the coding test from the WAIS-III battery is because that particular version of the test has been validated for fatigability in previous studies for other patient groups. See Moller et al, 2014. We have clarified the reason for this in the manuscript Page 10, lines 197-199

28) I am not familiar with the Ruff 2&7 and would value more comprehensive description of test and results. Similarly with the MapCog Spectra.

Response: While Ruff 2&7 is an established neuropsychological measure, please see the manual (Ruff RM, Allen CC. Ruff 2 & 7 Selective Attention Test. Lutz: Psychological Assessment Resources, Inc.; 1996), this, as you noted, is not the case with Mapcog Spectra. Thank you for noticing that. We have now added a more detailed description in the list of neuropsychological tests in the methods section. Page 10, lines 209-216 .

Table 1: Outcome measures.

29) The table would be clarified by being consistent with terminology. Is the fatiguing attention task in row 4 (Task fMRI) the same as the task in row 3, and the fatiguing vigilance task in row 5?

Response: Thank you for notifying this inconsistency. It is the same task and we have now clarified that in the table (page 11) by using e-prime vigilance task in row 3, 4, and 5.

30) In Row 4 of the secondary outcome measures should this refer to the WAIS-IV Digit Span? The left hand column suggests you are just looking at backward repetition, but the right hand column also refers to forward repetition. I recommend revising for clarity here, and justifying why you are selecting

Response: Thank you for noting this. We have now rewritten the text in the table so that the columns harmonize with each other Page 12, row 4.

31) Data management – please clarify if the data will be unidentifiable or reidentifiable?

Response: For clarity we have rephrased the sentence as follows: All data material will be recorded with a participant ID and will be unidentifiable for those who do not have access to the code-key. Page 13, lines 274-257.

Sample Size-

31) It is unclear which sample the power calculation is for, and the basis for using WAIS-III Coding test as each objective has different dependent variables?

The power calculation states that 24 is sufficient to detect a difference of 2.5 however, the sample sizes detailed in the participants section are 200 (neuropsychological assessments), 24 (fMRI) and 36 (blood sampling). Please justify the larger sample sizes for the first and third study component.

Response: Thank you for pointing this out. This Power calculation is also fully reasonable both for fMRI and blood sampling. We have clarified this in the text as follows “This size also holds for the fMRI sub study as an earlier study on mild traumatic brain injury has shown that 10 patients in each group was sufficient [ref]. The biomarkers data will be analyzed using multivariate data analysis (MVDA). Algorithms for sample size calculations using MVDA do not exist. MVDA is designed to handle a few subjects, low to variable ratios, and multiple intercorrelated variables. Based on our previous biomarkers study [ref], the sample size in this study will be sufficient to detect a difference between groups”. Page 14, lines 291-296.

Data analysis

32) On surface level the statistical plan seems reasonable however it is lacking in detail. Here I would expect to see a clear link between each objective listed, and an appropriate analysis planned, and recommend careful review.

Response: We have now gone through the statistical plan and added how we plan to analyze the data based on each study question. Please see the rewritten text. Page 14 - 15, lines 308 – 350.

33) Closing remarks: This protocol outlines a comprehensive body of work from Moller and colleagues on investigation into cognitive function and fatigue for patients with Chronic Pain. I recommend structural revisions to the introduction to improve the clarity and inclusion of additional details in the methods section to clarify the details and rationale.

Response: We have now made a comprehensive revision of the manuscript and considered your thoughtful comments. We hope that we have succeeded in clarifying the points need to be improved.

Reviewer: 3

Dr. David Carlson, VA Greater Los Angeles Healthcare System

Dear Dr Carlson,

Thank you for taking your valuable time to read and comment on our manuscript so carefully. We have answered your comments point by point below and have changed the manuscript accordingly. We now hope that you feel that the study protocol is clearer and can be accepted for publication.

Comments to the Author:

This is a protocol publication of a multi-component assessment of biometric, psychological, and neurophysiological variables in patients with chronic pain compared to normal controls.

- The study limitations are well-considered and comprehensive, and the reasons for the predominantly female sample is clearly laid out
- The references are appropriate and up-to-date, adequately supporting the context and relevance of the present study
- The psychological tests and scales used in the protocol are appropriate for the variables being studied, and many of the Swedish language instruments are described sufficiently to allow similar instruments to be used in other languages
- The methods are clearly and appropriately described

This review proposes only minor revisions to the submitted document:

1) - One concern exists regarding the exclusion criterion of "abuse." It would help to clarify the type(s) of abuse being referred to, as physical abuse (instead of emotional abuse or other forms) could have very different effects on the outcome variables being studied and complicate reproducibility of this study.

Response: Thank you for noting this. We refer to substance abuse, which is stated in a bullet point above. The bullet point, where it only says abuse has unfortunately been included by mistake and has now been deleted. Page 9, line 168.

The article is well-written in academic English, with only a few minor revisions to the grammar/spelling:

2) - The first line of the Abstract section should say "...presenting symptoms in health care and many patients with CP report..."

- The plural of analysis is "analyses" ("analyzes" is only used in American English for the third-person verb - e.g. "Thomas analyzes the information")
- put a comma after "of these" when it appears (e.g. "of these, 24 patients....")
- use "sensitivity to pain" instead of "sensibility for pain"
- under "Patients for Neuropsychological Assessment" section, write "... with CP from the Department of Rehabilitation Medicine at [name of institution] or the Department of Pain Rehabilitation at [Name of institution]..." ---- unless you're referring to inpatients who are "admitted to the..."
- The section after "Confidentiality" should be called "Data Analysis"

Response. Thank you so much for noting our typos. We have corrected these errors in the Abstract as well as in the main document. Page 2; Page 3, line 6; Page 7, lines 132, 133; Page 8, line 136; Page 15, line 360.

VERSION 2 – REVIEW

REVIEWER	Moore, David Liverpool John Moores University
REVIEW RETURNED	13-Feb-2023

GENERAL COMMENTS	The authors have attended to my comments, I have only one new comment, in the clarification that is present in the revision it appears that the authors report that the data has been collected here (see page 7). Assuming that this is true it seems somewhat problematic to present this as a protocol for review, as the protocol cannot at this stage be amended. In the research is complete I wonder if this is appropriate and if this should rather simply be submitted at a manuscript including data and discussion thereof for consideration.
---

REVIEWER	Alexander, Caroline Curtin University, School of Occupational Therapy, Sociak Work and Speech Pathology
REVIEW RETURNED	23-Feb-2023

GENERAL COMMENTS	I commend the authors on their revisions completed on this paper detailing the protocol for a valuable body of work on fatigue in patients with chronic pain. The introduction has increased clarity and reads well. The objectives are clear and specific, and the increased detail in the methodology and selected methodology is valued. As a final recommendation I suggest the authors consider consumer involvement in the preparation of outputs for dissemination including plain language summaries to enhance consumer understanding of this interesting work.
---

REVIEWER	Carlson, David VA Greater Los Angeles Healthcare System, Psychiatry
REVIEW RETURNED	13-Mar-2023

GENERAL COMMENTS	The revisions have been comprehensive and address the recommendations suggested by myself and other reviewers. The rationale for including predominantly female subjects, the handling of the appropriate imaging and questionnaire/measures. Areas of ambiguity are appropriately addressed.
---

VERSION 2 – AUTHOR RESPONSE

Reviewer: 1

Dr. David Moore, Liverpool John Moores University Comments to the Author:

The authors have attended to my comments, I have only one new comment, in the clarification that is present in the revision it appears that the authors report that the data has been collected here (see page 7). Assuming that this is true it seems somewhat problematic to present this as a protocol for review, as the protocol cannot at this stage be amended. In the research is complete I wonder if this is appropriate and if this should rather simply be submitted at a manuscript including data and discussion thereof for consideration.

Response to the reviewer: As the original paper was submitted before data collection was complete, and as the results have not yet been analyzed and reported, we still prefer to publish the protocol.

Reviewer: 2

Dr. Caroline Alexander, Curtin University Comments to the Author:

I commend the authors on their revisions completed on this paper detailing the protocol for a valuable body of work on fatigue in patients with chronic pain. The introduction has increased clarity and reads well. The objectives are clear and specific, and the increased detail in the methodology and selected methodology is valued.

As a final recommendation I suggest the authors consider consumer involvement in the preparation of outputs for dissemination including plain language summaries to enhance consumer understanding of this interesting work.

Response to the reviewer: Thank you for pointing out the importance of engaging consumers. We will consider this for future publications.

Reviewer: 3

Dr. David Carlson, VA Greater Los Angeles Healthcare System

Comments to the Author:

Areas of ambiguity are appropriately addressed.

Response to the reviewer: Thank you!